# The Effects of Obesity on Anti-Cancer Immunity and Cancer Immunotherapy

**DOI:** 10.3390/cancers12051230

**Published:** 2020-05-14

**Authors:** Matthew J. Woodall, Silke Neumann, Katrin Campbell, Sharon T. Pattison, Sarah L. Young

**Affiliations:** 1Department of Pathology, Dunedin School of Medicine, University of Otago, Dunedin 9054, New Zealand; wooma060@student.otago.ac.nz (M.J.W.); silke.neumann@otago.ac.nz (S.N.); katrin.campbell@otago.ac.nz (K.C.); 2Department of Medicine, Dunedin School of Medicine, University of Otago, Dunedin 9054, New Zealand; sharon.pattison@otago.ac.nz; 3School of Medical Sciences, Faculty of Medicine and Health, The University of Sydney, Sydney 2006, Australia

**Keywords:** obesity, metabolic syndrome, cancer, immunotherapy, checkpoint therapy, inflammation, T-cell exhaustion

## Abstract

Cancer is one of the leading causes of morbidity and mortality worldwide. Traditional treatments include surgery, chemotherapy and radiation therapy, and more recently targeted therapies including immunotherapy are becoming routine care for some cancers. Immunotherapy aims to upregulate the patient’s own immune system, enabling it to destroy cancerous cells. Obesity is a metabolic disorder characterized by significant weight that is an important contributor to many different diseases, including cancers. Obesity impacts the immune system and causes, among other things, a state of chronic low-grade inflammation. This is hypothesized to impact the efficacy of the immunotherapies. This review discusses the effects of obesity on the immune system and cancer immunotherapy, including the current evidence on the effect of obesity on immune checkpoint blockade, something which currently published reviews on this topic have not delved into. Data from several studies show that even though obesity causes a state of chronic low-grade inflammation with reductions in effector immune populations, it has a beneficial effect on patient survival following anti-PD-1/PD-L1 and anti-CTLA-4 treatment. However, research in this field is just emerging and further work is needed to expand our understanding of which cancer patients are likely to benefit from immunotherapy.

## 1. Introduction—Cancer and Obesity

Cancer is a non-communicable disease brought about by changes within cells, resulting in their uncontrolled growth and division [1]. It is one of the leading causes of mortality worldwide, and most deaths attributed to cancer worldwide are from lung, breast, colorectal, stomach and liver cancers [2,3]. Furthermore, the number of new cancer registrations is increasing globally [2].

Generally defined by a Body Mass Index (BMI) of ≥30, obesity is caused by an energy imbalance that favours weight gain, resulting in metabolic disturbances causing stress to tissues and ultimately leads to disease [4]. According to World Health Organisation estimates, in 2016, 39% of adults aged 18 years or older were found to be overweight and 13% were obese globally. The prevalence of these conditions has risen in both adults and children [2]. The increasing prevalence of obesity has resulted in increasing morbidity and years of life lost due to cardiovascular disease, type-2 diabetes mellitus, osteoarthritis, psychological problems and obesity-related cancers [5]. Reports from the World Cancer Research Fund and the International Agency for Research into Cancer have found that several types of cancer are associated with obesity, specifically endometrial, oesophageal adenocarcinoma, colorectal, breast cancer in postmenopausal women, prostate and renal cancers. Overall, the number of cases of cancer estimated to be caused by obesity is 20% and obesity is the second highest risk factor for cancer, after tobacco smoking [6]. Furthermore, studies have found that obesity leads to poorer cancer treatment efficacy and greater mortality from cancer [6,7,8,9]. Many factors are attributed to this, such as difficulties in adjusting dose for chemotherapy and positioning obese patients for radiation therapy [10,11]. Morbidly obese (BMI > 35) as well as underweight (BMI < 18.5) patients also have higher mortality rates following curative cancer resection surgery compared to normal weight and overweight patients [12].

## 2. Obesity and the Immune System

The links between obesity and cancer have largely been related to the effects of insulin resistance, elevated sex hormones, modulation of adipokine secretion, and upregulation of Programmed Cell Death Protein (PD)-1 expression [7,13]. Obesity affects the immune system in a way that is relevant in both cancer progression and treatment, and the general hypothesis is that these changes will reduce the efficacy of immune-based treatments. The key ways that obesity and the immune system interact are outlined below.

### 2.1. Chronic Inflammation

The obese state contributes to chronic inflammation by a number of mechanisms including adipocyte hypertrophy, macrophage recruitment and polarization, and increased production of pro-inflammatory mediators. Adipose tissue is required to expand in order to accommodate the influx of nutrients as seen in obesity [4]. In adults, adipocyte hypertrophy is favoured over hyperplasia [14]. These hypertrophic adipocytes induce shear mechanical stress on the extracellular environment and activate endoplasmic reticulum and mitochondrial stress responses. Overall, this results in a pro-inflammatory state within adipose tissue [15]. Figure 1 outlines the main contributors to this state.

The persistent state of inflammation and stress in adipose tissue leads to an increased expression of pro-apoptotic proteins, in particular Fas and its ligand, resulting in adipocyte cell death [16]. Following this, macrophages infiltrate the adipose tissue and encircle the dead adipocytes to form crown-like structures (CLS) [17,18]. CLS have been found in 50% of patients with breast cancer and their presence is associated with higher BMI and other systemic markers of metabolic syndrome. The formation of CLS causes the activation of pattern recognition receptors on macrophages, such as toll-like receptors (TLRs) [17]. As a result, macrophages are polarized towards a pro-inflammatory phenotype as opposed to an anti-inflammatory phenotype observed in healthy adipose tissue [19]. Other changes in the adipose tissue of obese individuals which drive inflammation include a reduction in the level of regulatory T-cells, increased fatty acid influx, vascularization, hypoxia, and increased leptin secretion [20]. Increases in proportions of neutrophils, dendritic cells (DCs), natural killer (NK) cells, mast cells, B-cells, T_h_1 CD4^+^ T-cells and CD8^+^ T-cells in the adipose tissue of obese individuals has also been shown [21]. One theory for this is an increased expression of Major Histocompatibility Complex (MHC)-II by adipocytes via a leptin-dependent mechanism, resulting in greater recruitment of leukocytes [21].

Both macrophages and hypertrophic adipocytes with increased intracellular stress upregulate secretion of the pro-inflammatory cytokines tumour necrosis factor (TNF)α, interleukin (IL)-1, IL-6, interferon (IFN)γ and monocyte chemoattractant protein-1 [22,23]. As well as promoting inflammation, these mediators also block the production of adiponectin, which has anti-inflammatory effects. This expression of cytokines is higher in the more pathogenic visceral adipose tissue compared to subcutaneous adipose tissue [24]. Furthermore, the production of anti-inflammatory cytokines such as IL-3, IL-4, IL-10 and IL-1 receptor antagonist is decreased [23]. Molecules such as TNFα are also pro-angiogenic, and support the development of tumours [25]. Overall, this results in the development of insulin resistance, increased lipolysis and impaired lipid storage [4]. This change towards a basal pro-inflammatory state in obesity has been identified beyond adipose tissue, including in leukocytes circulating in the blood of obese people. These cells demonstrate greater nuclear factor kappa-light-chain-enhancer of activated B cells (NF-κB) activation, which participates in the regulation of pro-inflammatory genes such as those involved in cytokine and chemokine production [26]. Furthermore, increased total lymphocyte, CD4^+^ and CD8^+^ T-cell and neutrophil counts have been associated with obesity [27,28,29]. However, several studies have failed to find consistent differences in the cytokine profile of non-obese and obese individuals that matches the current theories, although an increase in the pro-inflammatory cytokines IL-6 and TNFα have been observed [24,30,31,32]. This speaks to the complexity of the inflammatory changes brought about by obesity.

### 2.2. Altered Production of Immune Cells

Mobilization of fat stores as a result of increased lipolysis and impaired lipid storage in adipose tissue causes an accumulation of lipids in non-adipose tissue, including lymphoid tissues like the bone marrow and thymus [33]. Bone marrow-derived hematopoietic stem cells are continuously replicating in order to maintain lymphoid (T- and B-lymphocytes, NK cells) and myeloid-derived (monocytes, macrophages, DCs, granulocytes, erythrocytes, megakaryocytes, mast cells) lineages of cells [34]. Immature T-cells then travel to and undergo further development in the thymus. Increased lipid deposits in the thymus and bone marrow, both primary lymphoid organs, disrupt their integrity, altering the environment in which leukocytes develop [35]. In the bone marrow, this suppresses haematopoiesis and skews progenitor populations into producing a greater ratio of myeloid progenitor cells as opposed to lymphoid progenitor cells [36,37].

In the thymus, changes occur which resemble the natural process of thymic involution that normally occurs with aging [35]. This includes a loss of corticomedullary junctions, increased perithymic adiposity, and a reduction in populations of lymphocytic precursor cells [37]. These changes result in a reduced thymic output of naïve T-cells, which is likely to negatively affect immune surveillance and therefore increase the likelihood of immune escape of pathogens or tumours [37].

### 2.3. Reduction of T-Cell Variation

Obesity has been linked to a reduction in the diversity of T-cell receptors (TCRs) on circulating T-cells, reducing the number of antigens that can be recognized and responded to [33]. Obesity has also been shown to cause a reduction in lymph node size, impair lymphatic fluid transport and migration of DCs to peripheral lymph nodes, and reduce the number of T-cells in the lymph nodes. These changes reduce the ability of the immune system to recognize and effectively deal with foreign antigens [15]. Furthermore, the expansion of adipocytes caused by obesity suppresses anti-inflammatory pathways, enabling DCs and T-cells to become activated within visceral white adipose tissue [17]. However, constant presentation of antigens by DCs may eventually lead to T-cell exhaustion and chronic inflammation, reducing the capability for T-cells to have a successful effector response [17].

## 3. Cancer Immunotherapy

Immunotherapy is an attractive approach to treat cancer. Immunotherapy aims to harness the natural anti-tumour immune process and provide either passive or active immunity against cancers by stimulating the immune system to specifically target tumour cells. The formation of a memory response against cancer cells is then able to prevent a recurrence or relapse of the tumour [38]. Due to its specificity, immunotherapy has the potential to avoid some of the side effects of chemotherapy and radiation therapy [39]. There are several types of immunotherapy being used clinically and under development, including cancer vaccinations, adoptive cell transfer therapy, and immune checkpoint blockade [39].

Clinical trials have seen significant benefits of immunotherapies in some patients but not in others, and so identifying possible reasons that contribute to the variation in outcome is critical [40]. As outlined previously, the multiple effects of obesity on the immune system raises the potential that it may be a contributing factor for reduced efficacy of immunotherapeutic treatments.

### 3.1. Immune Checkpoint Molecules

One type of immunotherapy which has seen some success is the immune checkpoint blockade. Immune checkpoint molecules are receptors that are important in regulating the activation of the immune system. This is to ensure that expansion of cells in the adaptive wing of the immune system occurs only under the right conditions, preventing an inappropriate or overactive response [41].

Immune checkpoint molecules can either have a stimulatory or inhibitory effect. Stimulatory molecules include receptors such as cluster of differentiation (CD)28, CD80/86 and CD40 [42,43]. Following the internalization of antigen and subsequent migration to lymph nodes by antigen-presenting cells (APCs) such as DCs, CD80/86 receptors on DCs bind to CD28 on naïve T-cells and is required for the activation, differentiation and proliferation of T-cells into millions of clones [44]. Furthermore, CD4^+^ T-cells ‘license’ lymph node-resident DCs through CD40/CD40 ligand interaction, causing the upregulation of signals such as CD80/86 and the release of IFNγ and TNFα, which increase the activation and differentiation of CD8^+^ T-cells [45].

Important inhibitory immune checkpoint molecules are Cytotoxic T-lymphocyte-associated Protein (CTLA)-4 and Programmed Death (PD)-1. CTLA-4 is expressed by activated T-cells and a subset of regulatory T-cells, after the stimulation of CD28 on naïve T-cells during activation by DCs [46]. This molecule has a greater binding affinity with CD80/86 than CD28, thus acts as a competitive inhibitor by blocking a key part of T-cell stimulation [47]. Some evidence suggests that this binding may even produce inhibiting signals as well, further counteracting CD28-CD80/86 and TCR-MHC binding [47]. The relative levels of CD28 to CTLA-4 production therefore determines whether T-cell stimulation and proliferation occurs.

Another inhibitory checkpoint molecule is PD-1. This is expressed on the surface of T-cells, B-cells and Natural Killer (NK) cells, and binds to Programmed Death Ligand (PD-L)-1 and 2, which are upregulated on the surface of a wide variety of both immune and non-immune cells, including tumour cells [41]. The binding of PD-1 to its ligand causes similar effects as CTLA-4, inhibiting T-cell proliferation, reducing T-cell survival, and reducing the production of key cytokines such as IFNγ and TNFα. The key difference between these two molecules occurs around where in the T-cell response cycle they have their effect. CTLA-4 functions during the priming of immune cells, while PD-1 has its effect during the effector response [41].

Other inhibitory immune checkpoint molecules include lymphocyte activation gene (LAG)-3 (effects progression of the cell cycle) and T-cell immunoglobulin and mucin-domain-containing (Tim)-3, which inhibits the expression of cytokines such as TNF and IFNγ [48,49]. These are both present on the surface of T-cells. Because of the role that inhibitory checkpoint molecules play, they can be targeted to improve anti-cancer immunity.

### 3.2. Obesity and Immune Escape

Immune escape is one of the hallmarks of cancer development and is characterized by the immune system being unable to respond to and eliminate tumour cells. It is a complex process caused by tumour cells losing their antigenicity, losing their immunogenicity, the immunosuppressive nature of the tumour microenvironment (TME), or a combination of all three [50].

Tumour-infiltrating lymphocytes (TILs) are lymphocytes which have migrated into the tumour mass and surround or oppose the tumour cells [51]. They play an important role within the TME, either targeting tumours or inducing immunosuppression [52]. Because cancer is a chronic condition, prolonged antigen stimulation causes many of these cells to convert to an exhausted state [53]. This involves a decreased ability to cause an effector response, resulting in poorer control over cancers. Eventually, there is a depletion in the numbers of antigen-specific T-cells [49]. A hallmark of T-cell exhaustion is an increased expression of inhibitory checkpoint receptors on the cell surface such as PD-1, Tim-3 and LAG-3 [49].

Obesity has been shown to aid in immune escape by causing an increase in the frequency of exhausted T-cells. Peripheral T-cells in diet-induced obese (DIO) mice have increased expression of PD-1 compared to control mice. Both CD4^+^ and CD8^+^ T-cells, when stimulated ex vivo, displayed a reduced ability to proliferate and produce cytokines compared to T-cells from normal weight mice [54]. This was also observed in rhesus macaques and humans [54]. For instance, obese patients with melanoma show an increase in PD-1 expression within the TME, particularly in patients aged > 60, with increases in the expression of other markers of exhaustion as well: LAG-3, Tim-3, T cell immunoreceptor with immunoglobulin and immunoreceptor tyrosine-based inhibition motif domains (TIGIT), and the transcription factors T-Box Transcription Factor 21 (TBX21) and Eomesodermin (EOMES) [54]. The T-cells also had a lower proliferative potential [54]. Increased expression of PD-1 on TILs is associated with more aggressive features of cancer and poorer patient outcomes [55].

Similar observations have been made in other cancer types. An increased presence of PD-1^+^ CD8^+^ T-cells has been observed in breast tumours in high-fat diet mice compared to normal mice, which was associated with increased tumour progression [56]. However, tumour cells in obese mice did not have increased expression of the ligand for this receptor, PD-L1 [54]. This ligand is expressed by many cells, including tumour cells, and the binding of it to PD-1 on immune cells therefore aids in immune escape [57]. This finding means that increased activation of the PD-1/PD-L1 pathway, and consequent immunosuppression in obesity comes from increased PD-1 expression as opposed to PD-L1.

Wang et al. hypothesized that the increased expression of PD-1 in obese animals may be due to the raised levels of leptin as seen in obesity [54]. This hormone is produced by adipose tissue and acts to suppress food intake [58]. Many obese people develop leptin resistance, either due to the impaired transport of leptin across the blood–brain barrier, or problems downstream of the leptin signal. As a result, obese people often have high levels of leptin [59,60]. Signal transducer and activator of transcription (STAT)3, which is a major downstream transcription factor of the leptin receptor, can bind to the promoter region of PD-1 and cause the transcription and subsequent translation of this protein [61,62]. Stimulation of CD8^+^ T-cells by leptin has been shown to upregulate STAT3 and was correlated with increased expression of PD-1 [54]. Figure 2 shows a schematic representation of this hypothesis. Furthermore, increased levels of STAT3 have been associated with the expression of PD-1 in many cancers [54]. While elevated PD-1 expression is related to increased T-cell exhaustion and therefore reduced T-cell proliferation and function, it also means that therapies that target PD-1 have improved efficacy. This will be discussed in more detail in Section 4.

## 4. Obesity and Immune Checkpoint Blockade

Antibodies neutralising inhibitory immune checkpoint molecules typically have an anti-cancer effect through targeting PD-1/PD-L1 and CTLA-4 receptors. This is important as the expression of PD-L1 in numerous cancer types including head and neck squamous cell carcinoma, lung carcinomas, endometrial, ovarian, breast cancers and melanoma has been shown to contribute to the evasion by these tumour cells from the immune system [63]. CTLA-4 expression has been implicated in immune dysregulation of cervical, breast, lung gastric, colorectal, skin, non-Hodgkin’s lymphoma and B-cell chronic leukaemia [64]. Combination therapies against both of these receptors used in patients with melanoma and non-small-cell lung carcinoma have been effective at increasing tumour remission and survival, and trials for these treatments against other types of cancer are underway [65,66,67,68].

Because of the link between obesity and the immune system, there has been increasing interest over the past few years into analysing the effect that obesity has on immune checkpoint therapies. A retrospective study found that patients with a variety of cancers, including melanoma, non-small cell lung cancer, and renal cell carcinoma who are classified as overweight or obese have been shown to have a better response to anti-PD-1/PD-L1 immune checkpoint inhibitors [69,70]. This finding was also replicated in studies where melanoma or lung tumour-bearing DIO mice had an increased response to anti-PD-1 treatment compared to lean mice, with decreased melanoma metastases also being observed. The improved efficacy in DIO mice was associated with a significantly increased tumour-infiltrating T-cell count, increased CD8:CD4 ratio, and increased frequency of CD8^+^ T-cells in the TME. These factors are all considered to be correlated with positive outcomes [71,72,73]. The frequency of PD-1^+^ T-cells in the TME was also reduced in DIO mice after anti-PD-1 therapy, signifying an increased rate of T-cells which had been rescued from an exhausted state [74]. Increased expression of PD-1 by T-cells, is one of the potential reasons why obese mice and humans have stronger responses to anti-PD-1/PD-L1 treatment [74].

Several retrospective studies have confirmed the association of improved survival with increased weight in people with advanced/metastatic melanoma [75,76]. These studies found increased overall survival in overweight patients treated with either anti-PD-1/PD-L1 or anti-PD-1 + anti-CTLA-4 therapy. The association was predominantly found in males who had high serum creatinine levels (a marker for high muscle mass) [76]. Another study found that obese patients had a statistically significant improvement in progression-free survival (PFS) when treated with anti-PD-1 or anti-CTLA-4, although there was no improvement overall [77]. A separate study found a linear association between increased BMI and overall survival in patients with non-small cell lung cancer (NSCLC) treated with atezolizumab (anti-PD-L1). This association between BMI and survival was not found the control group who were treated with the chemotherapy agent, docetaxel. In particular, patients with a BMI ≥ 30 had significantly improved overall survival. Adverse events were not associated with differences in BMI [78]. Table 1 and Table 2 summarise studies investigating the effects of obesity on cancer immunotherapy outcomes.

Because obesity is a multi-faceted disease, it is likely that several pathways contribute to the observed clinical benefit of obesity on immune checkpoint blockade therapy. While no biological link has been confirmed yet, one proposed mechanism is that the increased expression of PD-1 triggered by heightened leptin levels is responsible for this phenomenon. It is conceivable that with increased expression of PD-1 on the surface of immune cells, the interaction between PD-1 and PD-L1 (on tumour cells) is increased, thus impairing anti-cancer immune responses. Anti-PD-1/PD-L1 therapy, which inhibits this interaction and allows CD8^+^ T-cells to have increased ability to kill tumour cells, would therefore be more efficacious (Figure 2). This theory is supported by the study by Kichenadasse et al., who found that the association between obesity and immune checkpoint blockade success was strongest in patients with a higher expression of PD-L1, while there was no difference in survival in patients with PD-L1 negative cancers [78]. This shows that checkpoint therapy can only be effective if the ligands for checkpoint molecules are expressed.

Fewer studies have looked at the effect of obesity on anti-CTLA-4 treatment. A retrospective study found that patients with metastatic melanoma, who were treated with ipilimumab as a monotherapy, had significantly increased response rates when patients had a BMI ≥ 25 (either overweight or obese) compared to those with a BMI < 25 (normal or underweight) [79]. No differences were found between gender or immune-related adverse effects. Overweight and obese patients also had a lower rate of brain metastases, and a trend of longer overall survival times. Another study also found a trend of increased overall survival and progression-free survival in obese males compared to normal weight males, but not females [75]. In contrast, a murine study looking at the effects of obesity on anti-CTLA-4 treatment of adenocarcinoma found reduced efficacy in obese BALB/c mice [80]. Lean mice and ob/ob (leptin deficient obese) mice had improved overall survival after anti-CTLA-4 therapy compared to the PBS control. However, DIO mice did not respond to treatment. One reason for this is thought to be the high levels of leptin in the DIO mice. When leptin levels were reduced, the ability to respond to treatment was returned, inferring a potential inverse relationship between leptin concentration and CTLA-4 expression. However, there is limited research into the link between leptin and CTLA-4 in the context of cancer and no potential mechanisms have been proposed.

One concern with administering immunotherapy in obese patients was the risk of an increase in immune-related adverse effects. For instance, it was found that the administration of anti-CD40/IL-2 immunotherapy into leptin-deficient obese mice resulted in a ‘cytokine storm’, with high levels of TNFα and IL-6 being released, causing multi-organ pathological responses and rapid lethality. This has been attributed to the baseline level of chronic inflammation found in the obese mice [81]. To date, this has rarely been observed in obese patients nor in preclinical mouse models [54,74,78]. One study did find that overweight and obese patients were twice as likely to suffer from immune-related adverse events, although not from higher grade adverse effects [70]. This study found, however, that an increase in these events was independently associated with increased efficacy of treatment [70].

## 5. Conclusions

Obesity creates a state of chronic inflammation, leading to reduction in the variation of T-cells, a polarization of macrophages towards a pro-inflammatory state, and the increased production of several inflammatory cytokines. This is widely thought to disturb immune-based therapies for diseases such as cancer, although studies around this subject are, on the whole, lacking. A surprising, but fairly consistent finding has been an increased efficacy of immune checkpoint blockade therapy for cancer in patients with increased BMI. The evidence is particularly strong for anti-PD-1/PD-L1 therapies, which has been investigated with more rigor, although evidence for anti-CTLA-4 therapies is pointing in this direction as well. The exact mechanisms behind this is unclear, however obesity causes an upregulation of markers of T-cell exhaustion including PD-1. Increased levels of leptin in particular have been linked to this observation.

Even though immunotherapies have markedly improved the survival for some people with cancer, many cancers do not respond. Therefore, it is vital to increase our understanding of the mechanisms driving resistance and response to these treatments, allowing better identification of which patients are most likely to benefit. Current research shows that obesity is likely to have an effect on treatment response and survival, and it is therefore important that further studies investigate this relationship. In particular, research into other forms of immunotherapy such as cancer vaccines and adoptive cell transfer therapy should be conducted to improve patient selection and outcomes.

## Figures and Tables

**Figure 1 cancers-12-01230-f001:**
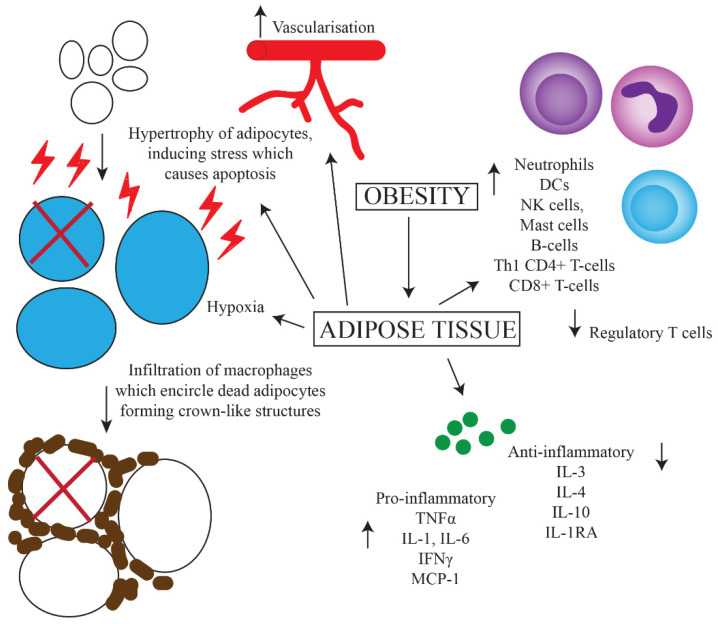
Predominant mechanisms of chronic inflammation caused by obesity. Increased uptake of nutrients leads to greater storage of fats and hence hypertrophy of adipocytes. This results in increased intracellular stress and upregulation of apoptotic genes, leading to apoptosis. Increased vascularization, hypoxia, cell death and upregulation of MHC-II on adipocytes leads to the influx of various inflammatory cells including macrophages, which surround dead adipocytes forming crown-like structures. There is also increased secretion of pro-inflammatory, and decreased secretion of anti-inflammatory cytokines.

**Figure 2 cancers-12-01230-f002:**
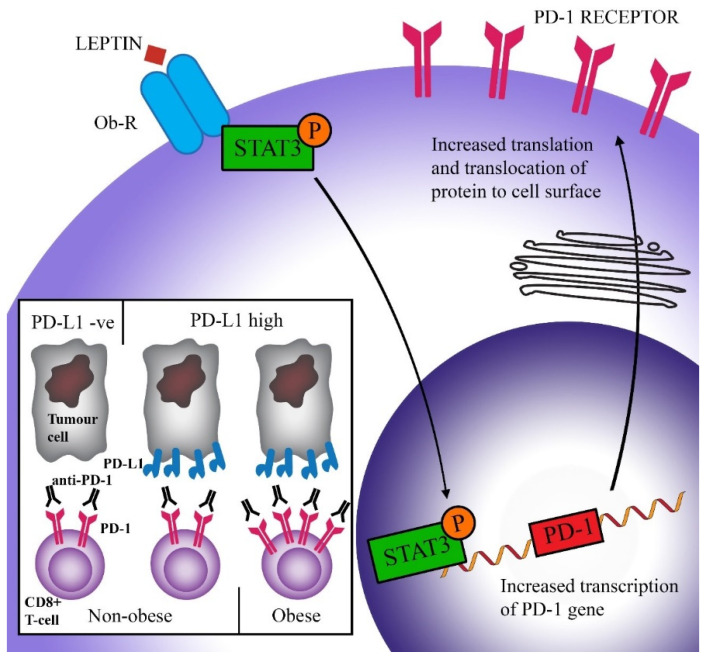
Scheme of a proposed pathway causing increased efficacy of immune checkpoint blockade in obese patients. Leptin binds to its receptor (Ob-R) on CD8^+^ T-cells and causes the activation (via phosphorylation) of the transcription factor STAT3. STAT3 triggers the transcription of the PD-1 gene and subsequent expression of the PD-1 protein on the cell surface. Higher PD-1 levels are correlated with increased exhaustion (activation by PD-L1 reduces T-cell proliferation, survival, and production of cytokines). However, increased PD-1 expression facilitates greater success of anti-PD-1 therapy, leading to increased overall survival in obese patients.

**Table 1 cancers-12-01230-t001:** Human studies published before February 2020 investigating the effects of obesity on immune checkpoint blockade therapy for cancer.

Study Authors	Date of Study	Type of Study	Cancer	Drug Name	Statistical Effects of Obesity
**Human trials**					
**Cortellini et al. [70]**	February 2019	Retrospective	NSCLC, melanoma, renal cell carcinoma, others	Anti-PD-1/PD-L1 (pembrolizumab, nivolumab or atezolizumab)	Objective response rate, time to treatment failure (HR = 0.51 [95% CI: 0.44–0.60], progress-free survival (HR = 0.46 [95%CI: 0.39–0.54]) and overall survival (HR = 0.33 [95%CI: 0.28–0.41]), significantly improved in overweight/obese patients (*p* < 0.0001)
**Donnelly et al. [77]**	August 2019	RCT	Metastatic melanoma	Anti-PD-1/anti-CTLA-4 (specific drugs unspecified)	No difference in PFS or OS between BMI levels in monotherapy however PFS for combination therapy was significant in obese patients (HR = 0.17 [95%CI: 0.04–0.65]) (*p* = 0.02)
**Kichenadasse et al. [78]**	December 2019	RCT	Non-small cell lung cancer	Atezolizumab (anti-PD-L1)	BMI ≥ 30 increase in OS (HR = 0.36 [95%CI: 0.21–0.62]) (*p* < 0.001)
**McQuade et al. [75]**	February 2018	Retrospective	Metastatic melanoma	Anti-PD-1/PD-L1, ipilimumab+ dacarbazine	Anti-PD-1/PD-L1: increased PFS (HR = 0.69 [95%CI: 0.45–1.06]) and OS (HR = 0.69 [95%CI: 0.42–1.12] for overweight and obese male patients compared to normal weight patients (not statistically significant), but not for female patients Anti-CTLA-4: increased PFS (HR = 0.55 [95%CI: 0.32–0.93]) and OS (HR = 0.40 [95%CI: 0.22–0.72]) in obese male patients compared to normal weight patients (not statistically significant), but not for female patients
**Naik et al. [76]**	March 2019	Retrospective	Unresectable or metastatic melanoma	Pembrolizumab or nivolumab (anti-PD-1) or anti-PD-1+ ipilimumab (anti-CTLA-4)	Overweight (but not obese) patients had increased OS compared to normal weight patients (HR = 0.26 [95%CI: 0.1–0.71]) (*p* = 0.038)
**Richtig et al. [79]**	October 2018	Retrospective	Metastatic melanoma	Anti-CTLA4 (ipilimumab)	Overweight and obese patients have higher response rates (*p* = 0.024, no other statistics provided) and a lower likelihood of brain metastases (8.6% vs. 32.5%, *p* = 0.012) compared to normal weight patients, as well as longer overall survival (HR = 1.81 [95%CI: 0.98–3.33], *p* = 0.056)
**Wang et al. [54]**	November 2018	RCT	Lung cancer, melanoma, ovarian cancer, and others (unspecified)	Anti-PD-L1/anti-PD-1 (specific drugs unspecified)	Improvement in progression free survival (median: 237 vs. 141 days, *p* = 0.0034) and overall survival (median: 523 vs. 361 days, *p* = 0.0492) in obese (BMI > 30) compared to non-obese (BMI < 30) patients

HR = hazard ratio, CI = confidence interval, OS = overall survival, PFS = progression-free survival.

**Table 2 cancers-12-01230-t002:** The effects of obesity on immune checkpoint blockade therapy for cancer trialled in animal studies.

Study Authors	Date of Study	Type of Study	Cancer	Drug Name	Statistical Effect of Obesity
**Animal trials**					
**Murphy et al. [80]**	August 2018	Tumour trial	Renca (renal adenocarcinoma)	Anti-CTLA-4	Compared to control, increased survival in lean mice (*p* = 0.007) and *ob*/*ob* mice (*p* = 0.005) but not DIO mice (*p* = 0.095), no other statistics provided
**Wang et al. [54]**	November 2018	Tumour trial	B16 (melanoma)	Anti-PD-1	DIO mice have reduced tumour growth by day 16 (*p* < 0.005), no other statistics provided
**Wang et al. [54]**	November 2018	Tumour trial	3ll (lung cancer)	Anti-PD-1	DIO mice have reduced tumour growth by day 11 (*p* < 0.001), no other statistics provided

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
