# Peer review of "The Effects of Obesity on Anti-Cancer Immunity and Cancer Immunotherapy"

_cancers, 2020, doi:10.3390/cancers12051230_

Round 1

Reviewer 1 Report

This review, “The effects of obesity on anti-cancer immunity and cancer immunotherapy” by Woodard et al provides a very nice summary of the interplay between obesity and host immunity and how checkpoint inhibition appears to be more successful in those that are obese, perhaps linked to altered check point marker expression during obesity. This information is timely given the evolution of checkpoint inhibition therapy strategies, and the presentation is straight forward.

Table 1 title is followed by abbreviations that are usually provided as a footnote

Reviewer 2 Report

This manuscript deals with a very important and interesting topic. Since a large proportion of the adult population is obese worldwide and obesity results in an altered immunity, both carcinogenesis and cancer immunotherapy can be affected and altered in obese patients.

The title of the manuscript is very promising but the review fails to provide a comprehensive and consistent synopsis of the topic. For a better understanding of this important issue, I suggest major revision of the submitted manuscript as follows:

  • Line 54: “Chronic inflammation” – a complex figure summarizing inflammatory processes related to obesity would be helpful
  • Lines 55-61: “chronic inflammation by a number of mechanisms”: how obese state is associated with chronic inflammation should be discussed more in detail on cellular (lymphocyte subsets involved) and molecular level (cytokines, chemokines etc.), locally and systemic changes e.g. cytokine levels in the blood. The number of cited papers supporting this theory is too little as well.
  • Line 62: as to my knowledge, apoptosis per se does not induce inflammation since apoptotic bodies are not immunogenic, please provide further information how enhanced apoptosis may trigger inflammatory responses.
  • Line 64 “crown-like structures” – please provide picture/figure
  • Line 67: “of macrophage pattern recognition receptors” is misleading please correct to pattern recognition receptors on macrophages.
  • Line 70: under which circumstances is an adipocyte able to produce proinflammatory cytokines? More information is needed here.
  • Line 73: inflammation and inflammatory cytokines are indirectly involved in carcinogenesis since they cannot directly act on DNA.
  • Lines 86-92: there are some incorrect information about the thymus. It is the site of T cell maturation taking up precursor cells from the bone-marrow. Furthermore, thymic involution starts already after puberty due to increased sex hormone levels and continues into old age.
  • Lines 113-141: this section does not discuss the topic of the paper at all and should be maximally shortened.
  • Lines 160-173: There is no need to describe stimulatory immune checkpoint molecules in detail, please shorten it.
  • Lines 178: “production of” should be replaced by “expression of”.
  • Lines 196-204: T-cell exhaustion is only a part of the immune status called immune escape during tumor development, please provide the complexity of the phenomenon.
  • Line 205: “Obesity and T-cell exhaustion” should be replaced by “Obesity and immune escape”. This should be the part of the paper where the effects of obesity on anti-cancer immunity will be discussed as mentioned in the title.
  • Line 290: “checkpoint molecules are expressed” should be completed as follows “ligands for checkpoint molecules are expressed”.
  • Lines 302-303: what is the association between leptin and CTLA-4?
  • Line 318: Table 2, please reformat it, leave more space for “date of study”

Reviewer 3 Report

In the manuscript entitled “The effects of obesity on anti-cancer immunity and cancer immunotherapy”, Woodall et. al review the effect of obesity on the immune system and immunotherapy. Obesity and immunotherapy are hot topics in cancer study, the review papers and new insights into these areas are welcome. Overall, this manuscript could be a good resource for learning the latest advances on the links between obesity and immune systems. However, the presentation is not easy to follow and distracted by too many texts away from the subjects. My comments are listed below:

  1. Some similar reviews published elsewhere should be considered such as, but not limited to, Aguilar et al, 2019 Curr Opin Immunol, PMID 29655021 and Deng et al, 2016 Annu Re Pathol, PMID: 27193454. It is also necessary to stress novel parts of this manuscript that distinguish it from other review papers in the area.
  2. The description of the cancer disease in abstract and introduction should be simplified. The authors should focus on the associations between obesity and cancer, and avoid clichés.
  3. The section “3. Current Cancer Treatments” is away from the key point of the manuscript and is suggested to be moved to the introduction part.
  4. There are some duplicated references, such as ref 13 and 26, 20 and 25, 28 and 29. It is required to check others.

Round 2

Reviewer 2 Report

The authors were very cooperative. Corrections and additions were carried out as suggested. Therefore, I suggest to accept the revised manuscript for publication in Cancers.

Reviewer 3 Report

The authors have addressed my concerns in the previous round of review.